# Diagnostic yield and the number of tumor cells of ultrathin bronchoscopy for peripheral lung lesions: A comparison with thin bronchoscopy

Atsuhiko Yatani[1], Naoko Katsurada[1], Takafumi Fukui[1], Jun Yamada[1], Hiroki Satoh[1], Chihiro Mimura[1], Daisuke Hazama[1], Masatsugu Yamamoto[1], Naoe Jimbo[2], Tomonori Tanaka[2], Motoko Tachihara[1]*

1 Division of Respiratory Medicine, Department of Internal Medicine, Kobe University Graduate School of Medicine, Kobe, Japan, 2 Department of Diagnostic Pathology, Kobe University Graduate School of Medicine, Kobe, Japan

* mt0318@med.kobe-u.ac.jp

**Data Availability Statement:** All relevant data are within the paper and its Supporting Information files.

## Abstract

Ultrathin bronchoscopy has been reported to have a higher diagnostic yield than thin bronchoscopy for small peripheral lung lesions in transbronchial biopsy under radial endobronchial ultrasonography (EBUS). However, data comparing the number of tumor cells in non-small cell lung cancer (NSCLC) are limited. We retrospectively compared the number of NSCLC tumor cells in peripheral lung lesions obtained using an ultrathin bronchoscope and a thin bronchoscope with radial EBUS between April 2020 and October 2021. In all patients, we used virtual bronchoscopic navigation (VBN) software, and guide sheaths were used in thin bronchoscopy cases. A total of 175 patients were enrolled in this study. Ultrathin bronchoscopy cases (n = 69) had lesions with a smaller diameter that are more peripherally located compared to thin bronchoscopy cases (n = 106) (median, 25.0 vs. 26.5 mm, mean bronchial generations accessed by bronchoscopy; 4.4±1.2 vs. 3.8±1.0, respectively; p<0.010). There were no significant differences in the overall diagnostic yield (ultrathin vs. thin bronchoscopy cases, 68.1% vs. 72.6%, p = 0.610) or diagnostic yield in only lung cancer cases (78.6% vs. 78.5%, p = 1.000). In histologically NSCLC cases (n = 102), the maximum number of tumor cells per slide as the primary endpoint was similar (average, 307.6 ±246.7 vs. 328.7±314.9, p = 0.710). The success rate of the Oncomine™ analysis did not differ significantly (80.0% vs. 55.6%, p = 0.247). The yield of NSCLC tumor cells was not different between the samples obtained by the ultrathin bronchoscope and those obtained by the thin bronchoscope.

## Introduction

Lung cancer has become the leading cause of cancer-related deaths worldwide [1], with an increasing number of cases and deaths, and treatment of lung cancer is becoming increasingly

**Funding:** The funders had no role in study design, data collection and analysis, decision to publish, or preparation of the manuscript.

**Competing interests:** The authors have declared that no competing interests exist.

important. New drugs targeting genetic mutations and immune checkpoint inhibitors have been introduced one after another, and patients eligible for such drugs experience a dramatically prolonged survival period. Lung cancer treatment guidelines recommend the selection of therapeutic agents based on the genetic mutation of the tumor tissue and programmed death ligand 1 (PD-L1) expression [2]; therefore, tumor tissue sampling is becoming increasingly important. In addition, next-generation sequencing (NGS) tests, which can detect multiple genetic abnormalities at once, are now being used in daily practice. The Oncomine™ Dx Target Test (ODxTT, Thermo Fisher Scientific, San Francisco, CA, USA) multi-CDx system is an amplicon-based hot spot panel test that analyzes genes using DNA and RNA derived from tumor samples [3]. In general, the amount of DNA required for a panel test is 10–500 ng [4]. The yield of DNA from one nucleated cell is estimated to be about 6 pg, and if 10 ng of DNA is obtained, extraction from approximately 2000 cells is required [5]. The number of tumor cells in the tissue to be analyzed is an important parameter.

Bronchoscopy has been devised in various ways to increase the diagnostic yield for lung cancer tissues. An ultrasound probe with a guide sheath (GS) is inserted close to the lesion through the forceps channel of the bronchoscope to confirm the location of the lesion by echo, and a biopsy was performed. This type of method, endobronchial ultrasonography (EBUS) with a GS (EBUS-GS), has been developed and shown to have good diagnostic yield and safety [6, 7]. Furthermore, the diagnostic yield for small peripheral pulmonary lesions increases when virtual bronchoscopic navigation (VBN) is combined with EBUS [8]. Ultrathin bronchoscopes (3.0 mm outer diameter) have smaller external diameters than thin bronchoscopes (4.0- or 4.2-mm outer diameter) and can reach more peripheral bronchi. In ultrathin bronchoscopy, the radial probe EBUS is directly inserted into the working channel without a GS. In contrast, a thin bronchoscope, in combination with GS, allows the location of peripheral lesions and repeat specimen collection through GS. Biopsy forceps of the same diameter were used in both cases. Ultrathin bronchoscopy has been reported to have a higher diagnostic yield than thin bronchoscopy for small peripheral lung lesions less than 30 mm in diameter in transbronchial biopsy under radial EBUS [9, 10]. However, the number of tumor cells in the specimens obtained was not compared between ultrathin and thin bronchoscopies. Therefore, we compared the number of tumor cells in non-small cell lung cancer (NSCLC) obtained by ultrathin bronchoscopy with that obtained by thin bronchoscopy using EBUS-GS methods.

## Material and methods

### Patients

We retrospectively compared the lesion characteristics, diagnostic yield, and number of tumor cells of NSCLC in peripheral lung lesions obtained using an ultrathin bronchoscope (MP290F, Olympus, Tokyo, Japan) and a thin bronchoscope (P290, P260F, Olympus, Tokyo, Japan) with EBUS from April 2020 to October 2021. Peripheral lung lesions were defined as those not visible on bronchoscopy. The eligibility criteria were patients aged 20 years or older, whose medical information could be obtained from the medical records, and who had undergone ultrathin or thin bronchoscopy for peripheral lung lesions. Patients who had requested not to participate in this study based on publicly available information were excluded. This study was approved by the Kobe University Ethics Committee (B210178) on September 27, 2021. This study was conducted in accordance with the principles of the Declaration of Helsinki. This study was registered in the University Medical Hospital Information Network in Japan (UMIN 000044777, https://center6.umin.ac.jp/cgi-open-bin/ctr/ctr_view.cgi?recptno=R000051144), with a registration date of September 27, 2021. From that point on, this study was initiated. Opt-out methods were used to obtain consent, which informed or disclosed

information about the conduct of the research, including the purpose of the research, and further ensures, to the extent possible, the opportunity to refuse. We had access to information that could identify individual participants during or after data collection by checking the medical records.

## Computed tomography evaluation of the lesion

We evaluated the lesions in terms of diameter, lobar location, location from the hilum, bronchus sign, and characteristics on computed tomography (CT). The lesion location from the hilum on CT images was classified into three groups: "inner" for lesions in the inner third ellipses, "middle" for lesions in the middle third ellipses, and "outer" for lesions in the outer third ellipse, and "middle" for lesions [11]. Bronchus sign was defined as the finding on cross-section of a bronchus leading directly to or contained within a nodule or mass [12]. The characteristics were classified as solid, part-solid, or ground-glass opacity nodules (GGN).

## Bronchoscopy method

All bronchoscopes and instruments were manufactured by Olympus (Tokyo, Japan). The ultrathin bronchoscope MP290F (3.0 mm bronchoscope diameter, 1.7 mm working channel diameter), and the thin bronchoscope P290 (4.2 mm bronchoscope diameter, 2.0 mm working channel diameter) or P260F (4.0 mm bronchoscope diameter, 2.0 mm working channel diameter) were entered through the planned bronchial route into the target lesion. Bronchoscopists decided the size of the bronchoscope to be used. Biopsy was performed using 1.5 mm diameter forceps (FB-233D) for transbronchial biopsy (TBB). Before bronchoscopy, we planned the bronchial pathway to the target lesions using virtual bronchoscopic navigation (VBN) software (Bf-NAVI®; Cybernet Systems, Tokyo, Japan), which automatically created virtual bronchial images from 1.0 mm slice width helical CT data. Bronchoscopy was performed by bronchoscopy specialists or supervised bronchoscopy residents under conscious sedation with intravenous midazolam, or midazolam and pethidine. For ultrathin bronchoscopy, a UM-S20-20R radial EBUS probe was inserted into the working channel once the bronchoscope reached close to the target lesion. If the lesion could be visualized by EBUS, the EBUS probe was withdrawn, and forceps (FB-233D) were inserted. Brushing twice and a target frequency of 10 biopsies was performed from the same lesion using the inserted forceps. For thin bronchoscopy, a UM-S20-20R radial EBUS probe was inserted through the GS into the working channel once the bronchoscope reached close to the target lesion. If the lesion could be visualized by EBUS, the EBUS probe was withdrawn, and forceps (FB-233D) were inserted through the GS. Forceps inserted through the GS were used to perform brushing twice, and 10 biopsies were obtained from the same lesion. Details of the bronchoscopes and forceps used are listed in Table 1, pictures are shown in Fig 1, and fluoroscopic images are shown in Fig 2.

**Table 1.  Details of the bronchoscopes used.**

|  | Ultrathin bronchoscope | Thin bronchoscope | |
| --- | --- | --- | --- |
|  | MP290F | P290 | P260F |
| Bronchoscope diameter | 3.0 mm | 4.2 mm | 4.0 mm |
| Working channel diameter | 1.7 mm | 2.0 mm | |
| Guide-sheath diameter | (-) | 1.9 mm | |
| Forceps diameter | 1.5 mm | 1.5 mm | |

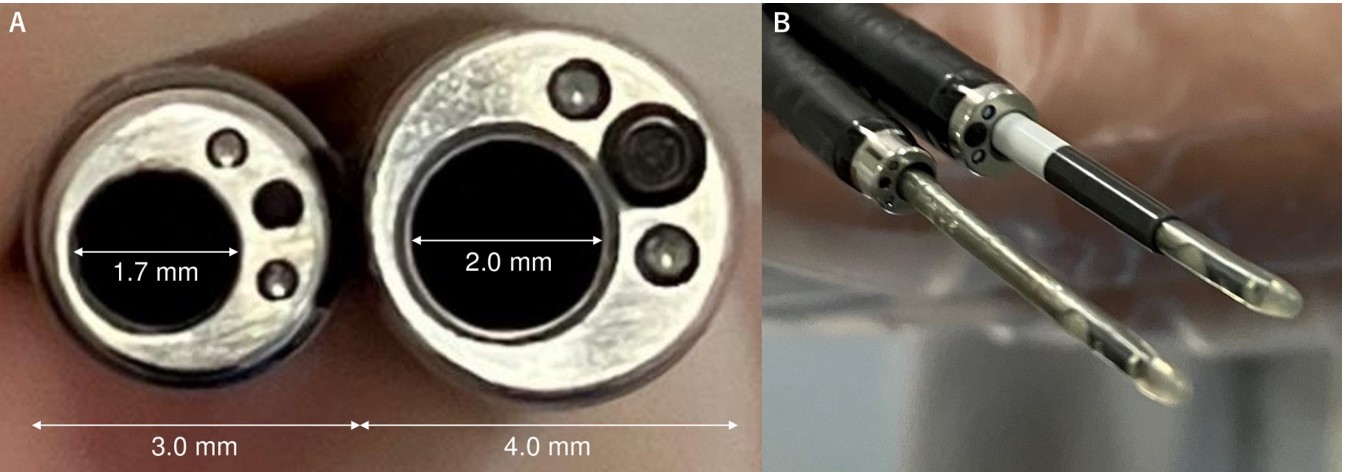

**Fig 1. Pictures of bronchoscopes.** A comparison of bronchoscopes. (A) The 3.0 mm ultrathin bronchoscope with a 1.7 mm working channel of MP290F (left), and the 4.0 mm thin bronchoscope with a 2.0 mm working channel of P260F (right). (B) MP290F with a radial EBUS probe (left), and P260F with a guide sheath and a radial EBUS probe (right).

## Pathological evaluation

The pathological diagnosis was confirmed by two pathologists by visually counting the number of cells based on hematoxylin-eosin (HE) staining.

## Endpoints

The primary endpoint was the maximum number of tumor cells per slide in the NSCLC cases. Secondary endpoints included diagnosis yield: percentage of all cases with a diagnosis made by

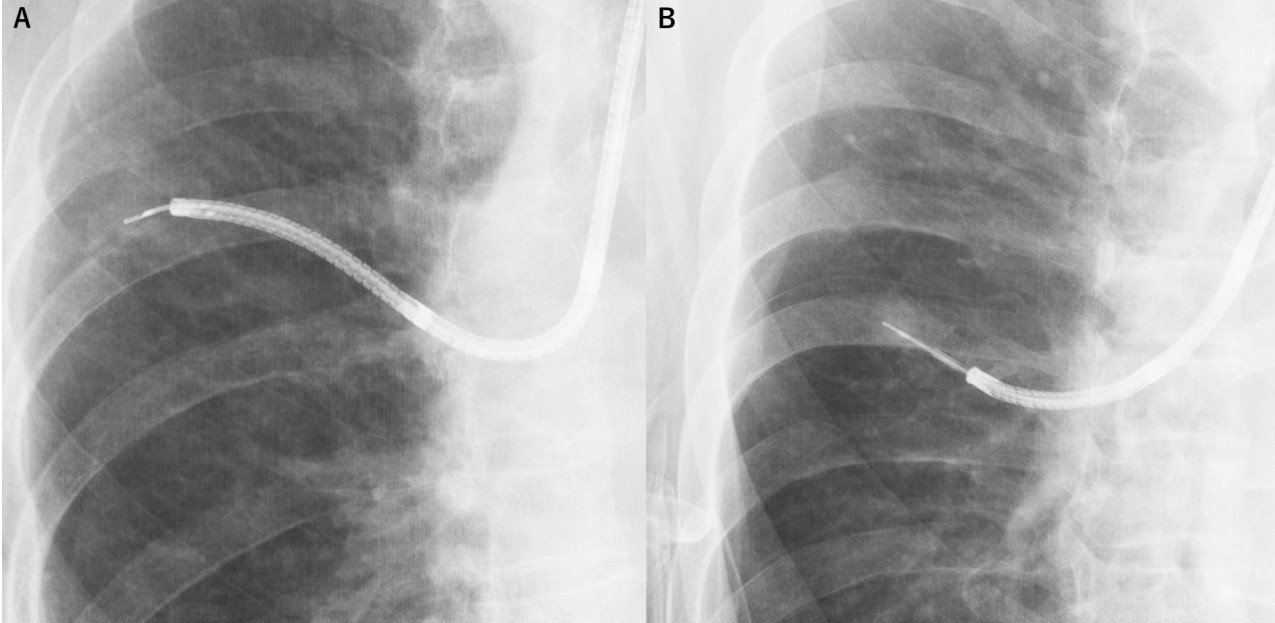

**Fig 2. Fluoroscopic images.** (A) Fluoroscopic image of transbronchial biopsy using the ultrathin bronchoscope of MP290F with the ultrathin bronchoscopic method. (B) Fluoroscopic image of transbronchial biopsy using the thin bronchoscope of P260F with a guide sheath method.

bronchoscopic tissue biopsy, histopathologic diagnosis, number of biopsies with a first diagnosis, number of biopsies in NSCLC cases in which tumor cells were first obtained by bronchoscopy, Oncomine™ analysis success rate, percentage of successful analyses for both DNA and RNA, and number of tumor cells per biopsy: average number of tumor cells per sample. If the lesion was not diagnosed, it was recommended that the patient undergo another diagnostic procedure such as CT-guided transthoracic needle aspiration biopsy (CTNB), retesting bronchoscopy, or surgery. We made a final diagnosis based on pathological evaluation or clinical follow-up.

### Data analyses

We created frequency tables for categorical variables and calculated summary statistics (number of cases, mean, standard deviation, minimum, median, and maximum) for continuous variables of patient background. To analyze the primary outcome, we summarized data on the median and range of maximum tumor cell counts between ultrathin and thin bronchoscopy cases. For analyses of the secondary outcomes, we summarized data on the diagnosis yield, histopathologic diagnosis, number of biopsies with a first diagnosis, Oncomine™ analysis success rate, and number of tumor cells per biopsy. We considered the p-value less than or equal to 0.050 to be statistically significant. We performed the Fisher's exact tests for qualitative data and t-tests for quantitative data. We also performed multiple regression analysis to evaluate factors affecting the number of tumor cells as the objective variable. We selected items considered clinically relevant and important as explanatory variables. All statistical analyses were performed using EZR version 3.6.3 (Saitama Medical Center, Jichi Medical University, Saitama, Japan), a graphical user interface for R (version 3.6.3; R Foundation for Statistical Computing, Vienna, Austria) [13].

## Results

### Consort flow chart

From April 2020 to October 2021, 175 patients were enrolled retrospectively in this study, with final analyses of 69 and 106 patients in the ultrathin bronchoscopy and thin bronchoscopy groups, respectively. Fig 3 shows the sorting flow chart.

### Characteristics of patients and lesions

The characteristics of the patients and their lesions are summarized in S1 Table. The target lesions in the ultrathin bronchoscopy cases were statistically significantly smaller and located more peripherally than those in the thin bronchoscopy cases (ultrathin vs. thin bronchoscopy cases, median (range), 25.0 (6.0–53.0) vs. 26.5 (9.0–104.0) mm, p<0.010; the mean bronchial generations accessed by bronchoscopy; 5.2±1.4 vs. 4.6±1.4, p<0.010). Bronchus sign was significantly more for ultrathin than thin lesions (bronchus sign present 67/69 (97.1%) vs. 79/106 (74.5%), p<0.010). In terms of shadow characteristics, solids were more common in thin bronchoscopy cases, and there were significant differences in all patterns (solid, 53/69 (76.8%) vs. 100/106 (94.3%); part solid, 13/69 (18.8%) vs. 6/106 (5.7%); GGN, 3/69 (4.3%) vs. 0/106 (0%), p<0.010).

### The number of tumor cells

We evaluated the number of tumor cells in samples obtained from 37 patients who were histologically diagnosed with NSCLC using ultrathin bronchoscopy and 63 patients using thin bronchoscopy. Table 2 shows the detailed biopsy results of histological NSCLC cases. There

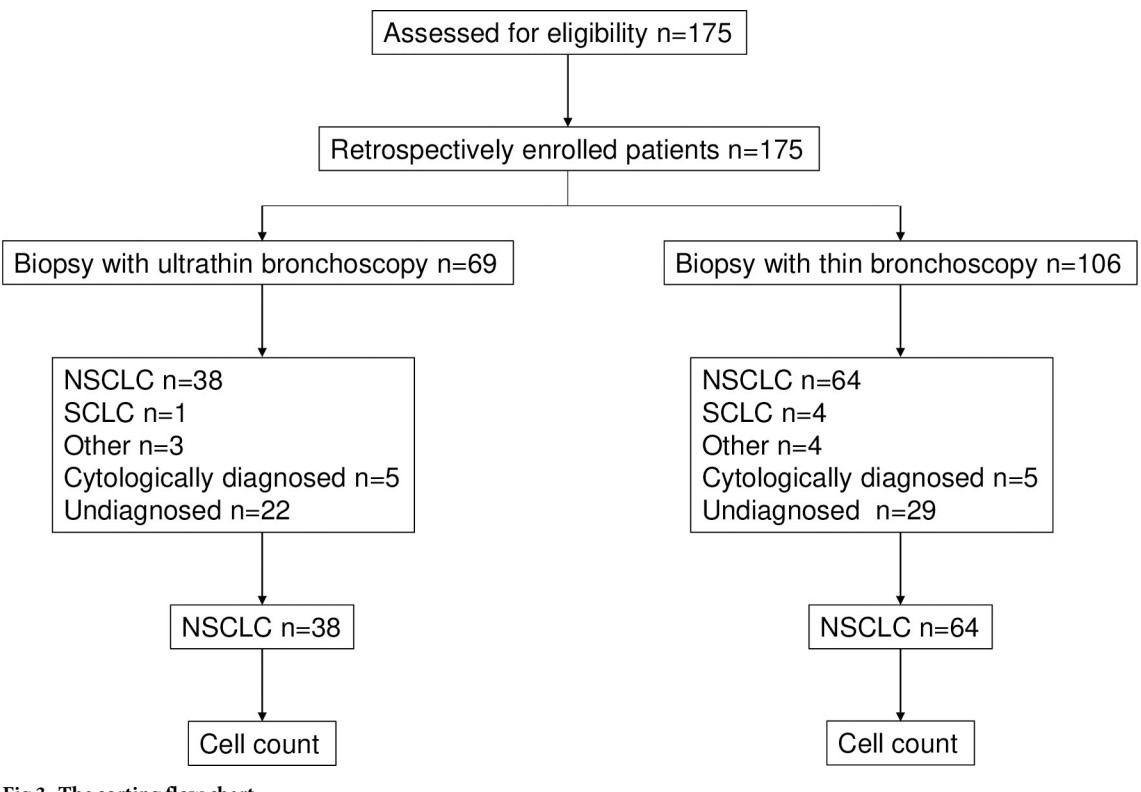

**Fig 3. The sorting flow chart.**

were no significant differences in the histological subtypes. The maximum number of tumor cells per slide was similar (average, 307.6±246.7 vs. 328.7±314.9, p = 0.710). The success rate of the Oncomine™ analysis did not differ significantly (80.0% vs. 55.6%, p = 0.247). There were slightly more cases of failed DNA analysis than RNA analysis in thin bronchoscopy.

Fig 4 shows the average number of tumor cells per biopsy in the two groups. The average number of tumor cells per biopsy did not decrease with the number of biopsies for either ultrathin or thin bronchoscopies. The number of specimens with the first diagnosis was similar (median: 2 (1–9) vs. 2 (1–12), p = 0.355).

**Table 2. The detailed biopsy results of histologically NSCLC cases.**

| | Ultrathin bronchoscopy (n = 38) | Thin bronchoscopy (n = 64) | p-value |
|---|---|---|---|
| | n (%) | n (%) | |
| Histological diagnosis | | | 0.849 |
| Adenocarcinoma | 23 (62.2) | 36 (57.1) | |
| Squamous cell carcinoma | 12 (32.4) | 22 (34.9) | |
| Non-small cell carcinoma | 2 (5.4) | 5 (7.9) | |
| Maximum number of tumor cells (mean±SD, n = 100) | 307.6±246.7 | 328.7±314.9 | 0.710 |
| Number of biopsies (median; range) | 9 (5–12) | 9 (1–13) | 0.622 |
| Specimen number with the first diagnosis (median; range) | 2 (1–9) | 2 (1–12) | 0.355 |
| The success rate of Oncomine™ analysis | 8/10 (80.0) | 10/18 (55.6) | 0.247 |
| DNA analysis | 9/10 (90.0) | 10/18 (55.6) | 0.098 |
| RNA analysis | 8/10 (80.0) | 13/18 (72.2) | 1.000 |

Maximum number of tumor cells: number of tumor cells in the specimens with the highest tumor content.

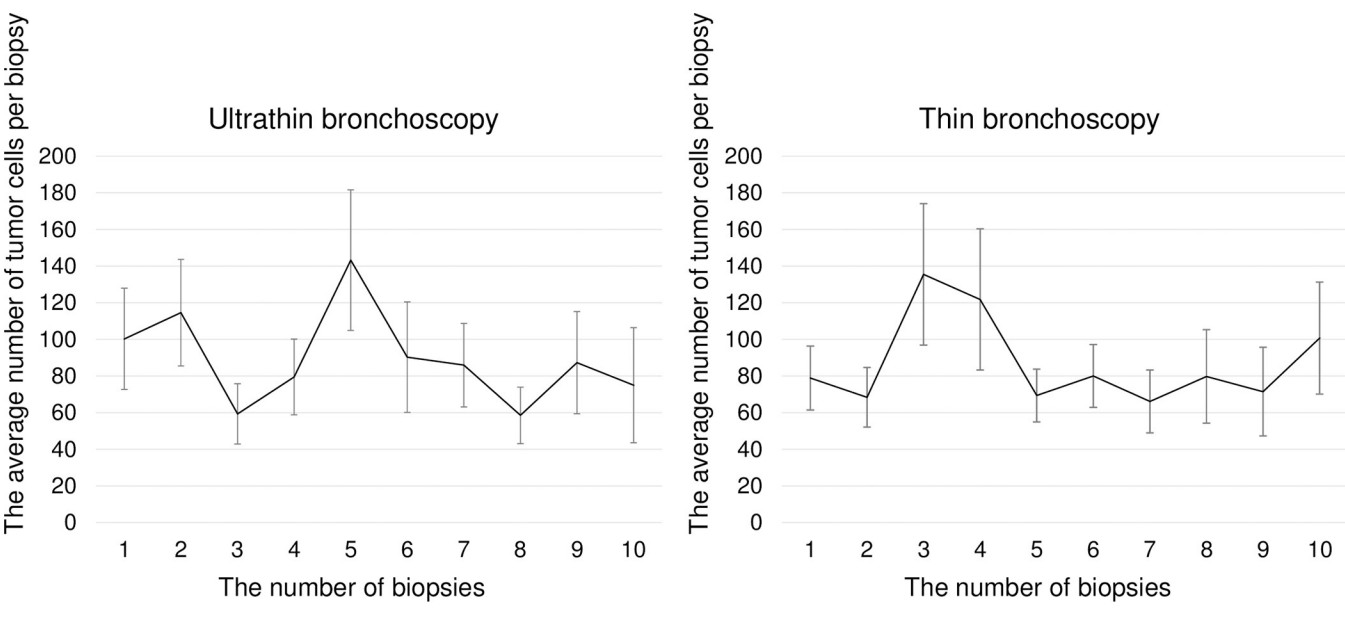

**Fig 4. The average number of tumor cells per biopsy in the two groups.**

We performed a multiple regression analysis with the number of tumor cells as the objective variable and bronchoscope selection, lesion size, bronchus generation on VBN, and histological diagnosis as explanatory variables and found that bronchus generation on VBN which meant how peripherally located the lesion was and histological diagnosis could affect the number of tumor cells even when corrected for bronchoscope selection and lesion size. Table 3 shows the result.

## Diagnostic yield

The diagnostic yield was defined as the percentage of cases in which a tissue biopsy was attempted from the lesion by bronchoscopy and some definitive tissue or cellular diagnosis was obtained. The overall diagnostic yield was 68.1% (47/69) in 69 patients who underwent a biopsy with an ultrathin bronchoscope. Thirty-eight patients were diagnosed with NSCLC, one with small cell lung cancer (SCLC), and three with other diagnoses based on the histological biopsy samples. Five patients were diagnosed with lung cancer based on cytological analysis alone. Among the 22 patients undiagnosed by ultrathin bronchoscopy, 11 were diagnosed with lung cancer by TBNA, CTNB, and surgery. One patient was diagnosed with lung cancer. Ultimately, 56 patients were diagnosed with lung cancer. Therefore, the diagnostic yield for lung cancer using ultrathin bronchoscopy was 78.6% (44/56). In contrast, of the 106 patients who underwent a biopsy with a thin bronchoscope, the overall diagnostic yield of thin bronchoscopy was 72.6% (77/106). Sixty-four patients were diagnosed with NSCLC, four with SCLC,

**Table 3. The result of a multiple regression analysis of the number of tumor cells.**

|  | β | SE | t-value | p-value |
| --- | --- | --- | --- | --- |
| Bronchoscope selection | -0.026 | 59.653 | -0.262 | 0.794 |
| Lesion size | -0.033 | 2.071 | -0.332 | 0.741 |
| Bronchus generation on VBN | -0.280 | 18.896 | -2.827 | 0.006 |
| Histological diagnosis | 0.195 | 46.078 | 2.014 | 0.047 |

**Table 4. Characteristics of cases in which the final diagnosis of lung cancer was made by bronchoscopy.**

| | Ultrathin bronchoscopy (n = 56) | Thin bronchoscopy (n = 93) | p-value |
|---|---|---|---|
| | n (%) | n (%) | |
| Lesion size (mm, median; range) | 25.5 (6–53) | 28.0 (9–80) | 0.017 |
| Lobar location | | | 0.546 |
| Right upper lobe | 15/18 (83.3) | 23/26 (88.5) | |
| Right middle lobe | 2/3 (66.7) | 5/8 (62.5) | |
| Right lower lobe | 10/12 (83.3) | 11/15 (73.3) | |
| Left upper lobe | 9/14 (64.3) | 24/30 (80.0) | |
| Left lower lobe | 8/9 (88.9) | 10/14 (71.4) | |
| Lesion location from the hilum on CT images | | | <0.010 |
| Inner | 3/3 (100.0) | 16/16 (100.0) | |
| Middle | 8/10 (80.0) | 24/34 (70.6) | |
| Outer | 33/43 (76.7) | 33/43 (76.7) | |
| Bronchus sign | | | 0.028 |
| Present | 42/54 (77.8) | 58/69 (84.1) | |
| Absent | 2/2 (100.0) | 15/24 (62.5) | |
| Characteristics | | | <0.010 |
| Solid | 35/42 (83.3) | 70/87 (80.5) | |
| Part solid | 9/11 (81.8) | 3/6 (50.0) | |
| GGN | 0/3 (0.0) | - | |
| Bronchus generation on VBN (mean±SD) | 5.3±1.4 | 4.5±1.5 | <0.010 |

and four with other diagnoses based on histological biopsy samples. Five patients were diagnosed with lung cancer based on cytological analysis alone. Among the 29 patients undiagnosed by thin bronchoscopy, 16 were diagnosed with lung cancer by TBNA, CTNB, and surgery. Four patients were diagnosed with lung cancer. As a result, 93 patients were ultimately diagnosed with lung cancer. Therefore, the diagnostic yield of lung cancer by thin bronchoscopy was 78.5% (73/93). Table 4 shows characteristics of cases in which the final diagnosis of lung cancer was made by bronchoscopy. Ultrathin bronchoscopy cases (n = 56) had lesions with a smaller diameter and more peripherally located on virtual bronchoscopy than thin bronchoscopy cases (n = 93) (median (range), 25.5 (6.0–53.0) vs. 28.0 (9.0–80.0) mm, p = 0.02, the mean bronchial generations accessed by bronchoscopy; 5.3±1.4 vs. 4.5±1.5, p<0.010). There were no significant differences in the overall diagnostic yield or the diagnostic yield in lung cancer cases (78.6% vs. 78.5%, p = 1.000).

## Discussion

Our study showed that the number of tumor cells and diagnostic yield were similar between ultrathin and thin bronchoscopy, despite the fact that ultrathin bronchoscopy was selected for smaller tumors with a greater number of bronchi. The reason for the similar number of tumor cells obtained with ultrathin and thin bronchoscopy may be that the forceps diameter used was the same. The results of a multiple regression analysis we performed in this study showed that bronchus generation on VBN which meant how peripherally located the lesion was and histological diagnosis could affect the number of tumor cells even when corrected for bronchoscope selection and lesion size. In other words, bronchoscope selection had no effect on the number of tumor cells. We consider peripheral lesions to be difficult to tension and the number of tumor cells obtained to be low. In the case of ultrathin bronchoscopy, there is a possibility that the forceps do not reach the lesion, although they appear to have reached the lesion

under fluoroscopy, or the specimens obtained occasionally disappear during specimen preparation because of their small size [14]. As the average number of biopsies in this study was nine, it was considered sufficient. In a previous study, the number of tumor cells decreased gradually as the number of biopsies by thin bronchoscopy increased, suggesting that local bleeding may be the cause [15]. In the present study, the number of cells did not gradually decrease, which may be because the biopsy was performed at the optimal site by reconfirming the lesion site by EBUS during the biopsy.

In a previous study, the number of tumor cells was smaller and the success rate of ODxTT was significantly lower [16]. It is necessary to collect sufficient tumor cells to proceed with the genetic testing and other tests necessary for treatment selection. In the present study, the success rate of ODxTT was not significantly different between ultrathin and thin bronchoscopy cases, although the number of cases of ODxTT was limited. This is presumably because the number of tumor cells was similar in both cases. In a previous report, the success rate of ODxTT in the group that performed TBB using only small forceps was 70% [17]. In the present study, the success rate of the ODxTT in the ultrathin bronchoscopy cases was not inferior, and it was possible to collect specimens suitable for ODxTT. Therefore, it may be useful to perform ODxTT even in cases with ultrathin bronchoscopy. There were slightly more cases of failed DNA analysis than RNA analysis in thin bronchoscopy. Possible reasons for DNA failure included insufficient specimen volume, smaller tissue surface area, and over-fixation. Therefore, it is necessary to submit more specimens or to prevent over-fixation.

We were able to perform a sufficient biopsy using the ultrathin bronchoscope instead of EBUS-GS. The ultrathin bronchoscope of MP290 used in the present study has a thin diameter of 3.0 mm, which allows it to reach a peripheral lesion. In addition, it has a 1.7 mm working channel, which allows the use of a radial probe EBUS and 1.5 mm biopsy forceps. In a previous study, it was reported that the mean bronchus generation reached with the ultrathin bronchoscope, thin bronchoscope, and VBN was 5.5, 4.4, and 5.1, respectively [10]. Another study reported that the ultrathin bronchoscope could reach more distal bronchi than the thin bronchoscope (median, fifth-generation bronchi in the UTB group, and fourth-generation bronchi in the TB-GS group; P < 0.001) [9]. In fact, in the present study, the mean bronchus generation reached with the ultrathin bronchoscope was greater than that with the thin bronchoscope (mean±SD, 4.4±1.2 vs. 3.8±1.0, p<0.010). Thus, we were able to reach similar peripheral lesions as previously reported [9, 10]. In a previous study, the diagnostic yield of flexible bronchoscopy used to evaluate peripheral lesions less than 30 mm in diameter was 70.1% using ultrathin bronchoscopy and 58.7% using thin bronchoscopy [10]. In the present study, the overall diagnostic yield was 68.1% in the ultrathin bronchoscopy cases and 72.6% in the thin bronchoscopy cases; therefore, the diagnostic yield did not differ significantly from that reported previously. The present study was retrospective, and an ultrathin bronchoscope was chosen for more peripheral lesions in the hope that it would be more selective for more peripheral bronchi. For lesions that require more peripheral bronchi to be selected, ultrathin bronchoscopes may be appropriate based on their high diagnostic yield and genetic analysis success rates.

Our study had several limitations. First, this study was a retrospective comparison of ultrathin and thin bronchoscopy cases, not a randomized trial, and the targets of the two groups were not aligned. In addition, subjective bias may have existed because the choice of the bronchoscope to use for the target lesion was the bronchoscopist's decision. Second, the cell count was performed by a single pathologist, which may have also included a subjective element. Third, the number of ODxTT submissions were limited. This was because we scheduled to submit them to surgical specimens if the lung cancers were in operable staging. Fourth, this study was conducted at a single institution and had a small sample size. In particular, since the number of cases with genetic analyses was small, further research is required.

In conclusion, our study indicated that the number of tumor cells and diagnostic yield were similar between ultrathin and thin bronchoscopy, even for smaller tumors with a large number of bronchi in ultrathin bronchoscopy cases. Therefore, it is important to choose an appropriate bronchoscope depending on the location and size of the lesion, and an ultrathin bronchoscope may be a better choice for small peripheral lesions.

## Supporting information

**S1 Checklist. STROBE Statement—checklist of items that should be included in reports of observational studies.**
(DOCX)

**S1 Table. The characteristics of the patients and their lesions.**
(DOCX)

**S1 Data.**
(XLSX)

## Acknowledgments

The authors are grateful to all participating patients, their families, and medical staff.

## Author Contributions

**Conceptualization:** Naoko Katsurada, Motoko Tachihara.

**Data curation:** Atsuhiko Yatani, Naoko Katsurada, Takafumi Fukui, Jun Yamada, Hiroki Satoh, Chihiro Mimura, Daisuke Hazama, Masatsugu Yamamoto, Naoe Jimbo, Tomonori Tanaka, Motoko Tachihara.

**Formal analysis:** Atsuhiko Yatani, Naoko Katsurada, Takafumi Fukui, Jun Yamada, Hiroki Satoh, Chihiro Mimura, Daisuke Hazama, Masatsugu Yamamoto, Naoe Jimbo, Tomonori Tanaka, Motoko Tachihara.

**Writing – original draft:** Atsuhiko Yatani.

**Writing – review & editing:** Atsuhiko Yatani, Naoko Katsurada, Takafumi Fukui, Jun Yamada, Hiroki Satoh, Chihiro Mimura, Daisuke Hazama, Masatsugu Yamamoto, Naoe Jimbo, Tomonori Tanaka, Motoko Tachihara.

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
