## [Decision Letter · Decision Letter 0]

13 Jun 2023

PONE-D-23-12382Diagnostic yield and the number of tumor cells of ultrathin bronchoscopy for peripheral lung lesions: a comparison with thin bronchoscopyPLOS ONE

Dear Dr. Tachihara,

Thank you for submitting your manuscript to PLOS ONE. After careful consideration, we feel that it has merit but does not fully meet PLOS ONE’s publication criteria as it currently stands. Therefore, we invite you to submit a revised version of the manuscript that addresses the points raised during the review process.

ACADEMIC EDITOR: The reviewers have recommended publication, but also suggest some minor revisions to your manuscript.  Therefore, I invite you to respond to the reviewers' comments and revise your manuscript.

We look forward to receiving your revised manuscript.

Kind regards,

Fumihiro Yamaguchi

Academic Editor

PLOS ONE

Journal Requirements:

Reviewers' comments:

Reviewer's Responses to Questions

**Comments to the Author**

1. Is the manuscript technically sound, and do the data support the conclusions?

Reviewer #1: No

Reviewer #2: Yes

Reviewer #3: Yes

2. Has the statistical analysis been performed appropriately and rigorously? 

Reviewer #1: N/A

Reviewer #2: Yes

Reviewer #3: Yes

3. Have the authors made all data underlying the findings in their manuscript fully available?

Reviewer #1: No

Reviewer #2: Yes

Reviewer #3: Yes

4. Is the manuscript presented in an intelligible fashion and written in standard English?

Reviewer #1: Yes

Reviewer #2: Yes

Reviewer #3: Yes

5. Review Comments to the Author

Reviewer #1: A study designed with an effort to reach and diagnose peripheral lung nodules

There are serious methodology errors in the study.

Low number of patients

It is mentioned that the lesions are peripheral, but there is no classification for this.

Which lesion and how much peripheral is unclear?

There was no information on how far the lungs could go with the ultra-thin bronchoscope.

Results and comparison groups are unclear

It is not appropriate for the study to be published in a SCI journal as it is.

Reviewer #2: The manuscript of Yatani and colleagues aims to compare the diagnostic yield and number of tumor cells of ultrathin bronchoscopy vs. thin bronchoscopy for peripheral lung lesions. With the recent advent and promises of ultrathin bronchoscopy, the results of the current study are of interest for pulmonologists. The current manuscript is well written in a synthetic manner. The study is limited by its retrospective nature.

I would have the following comments:

1. The diagnostic yield is the primary endpoint of the study and should be defined more precisely.

2. The current study is retrospective. Did the author try to adjust/compensate for potential selection biases? Only crude statistical comparisons are reported in the Results section. Multivariate statistical analyses could be used.

3. Please report the p-values with at least 3 digits.

4. Line 219, what does the symbol Â stand for? A table of abbreviations would be useful.

5. Table 3, first line, one should read "Diagnostic yield" instead of Diagnosis yield.

6. If Figure 1 and 2 are independent, Figure 2 left and right panels should be read A and B (instead of C and D).

Reviewer #3: 1. The PI and team can provide additional analysis by also using ANOVA for comparisons of the bronchoscopy. If the assumptions of ANOVA are not meant, The PI and team can go ahead and use non parametrics, especially the Kruskal-Wallis test which compares the median. The team can even use Welch's ANOVA which adjusts for unequal variances and provides reliable results even when the assumption is not met.

2. If possible, the PI and team can cite current literature (> 10years) and remove the old literature as seen in references, page 22 to page 23 lines 369-370, line 372-374, and line 380-381.

6. PLOS authors have the option to publish the peer review history of their article (what does this mean?). If published, this will include your full peer review and any attached files.

Reviewer #1: No

Reviewer #2: No

Reviewer #3: **Yes: **RICHARD ATUGONZA

---

## [Author Response · Author response to Decision Letter 0]

24 Jul 2023

We thank the reviewers for their insightful comments, which we have now addressed in both the revised paper and the attached point-by-point responses. We hereby submit the revised version of our manuscript, which we hope is now acceptable for publication in PLOS ONE. We are looking forward to your favorable consideration.

---

## [Decision Letter · Decision Letter 1]

14 Aug 2023

Diagnostic yield and the number of tumor cells of ultrathin bronchoscopy for peripheral lung lesions: a comparison with thin bronchoscopy

PONE-D-23-12382R1

Dear Dr. Tachihara,

We’re pleased to inform you that your manuscript has been judged scientifically suitable for publication and will be formally accepted for publication once it meets all outstanding technical requirements.

Kind regards,

Fumihiro Yamaguchi

Academic Editor

PLOS ONE

Additional Editor Comments (optional):

Reviewers' comments:

Reviewer's Responses to Questions

**Comments to the Author**

1. If the authors have adequately addressed your comments raised in a previous round of review and you feel that this manuscript is now acceptable for publication, you may indicate that here to bypass the “Comments to the Author” section, enter your conflict of interest statement in the “Confidential to Editor” section, and submit your "Accept" recommendation.

Reviewer #2: All comments have been addressed

2. Is the manuscript technically sound, and do the data support the conclusions?

Reviewer #2: Yes

3. Has the statistical analysis been performed appropriately and rigorously? 

Reviewer #2: Yes

4. Have the authors made all data underlying the findings in their manuscript fully available?

Reviewer #2: Yes

5. Is the manuscript presented in an intelligible fashion and written in standard English?

Reviewer #2: Yes

6. Review Comments to the Author

Reviewer #2: All my comments have been addressed in an appropriate manner. The quality of the manuscript improved significantly after revision.

7. PLOS authors have the option to publish the peer review history of their article (what does this mean?). If published, this will include your full peer review and any attached files.

Reviewer #2: No

---

## [Editor Report · Acceptance letter]

17 Aug 2023

PONE-D-23-12382R1 

Diagnostic yield and the number of tumor cells of ultrathin bronchoscopy for peripheral lung lesions: a comparison with thin bronchoscopy 

Dear Dr. Tachihara:

I'm pleased to inform you that your manuscript has been deemed suitable for publication in PLOS ONE. Congratulations! Your manuscript is now with our production department. 

Kind regards, 

on behalf of

Dr. Fumihiro Yamaguchi 

Academic Editor

PLOS ONE